# Usability and Clinical Performance Characteristics of the Asante HIV1/2 Test by Trained Users in Two African Sites

**DOI:** 10.3390/diagnostics11091727

**Published:** 2021-09-20

**Authors:** Mohammed Majam, Naleni Rhagnath, Vanessa Msolomba, Leanne Singh, Michael S. Urdea, Samanta T. Lalla-Edward

**Affiliations:** 1Ezintsha, Faculty of Health Sciences, University of the Witwatersrand, Johannesburg 2193, South Africa; mmajam@ezintsha.org (M.M.); nrhagnath@ezintsha.org (N.R.); vmsolomba@ezintsha.org (V.M.); thewritepages@gmail.com (L.S.); 2Halteres Associates, 2010 Crow Canyon Place, Suite 100, San Ramon, CA 94583, USA; murdea@halteresassociates.com

**Keywords:** sensitivity, specificity, HIV counselling and testing, rapid diagnostic test, point-of-care

## Abstract

HIV self-testing (HIVST) devices are acknowledged as having the potential to enable the acceleration of HIV diagnosis and linkage to care. How efficiently professional and trained users engaged with the Asante HIV-1/2 Oral Self-Test (Asante) (Sedia Biosciences, Portland, OR, USA), and the accuracy of the device in comparison to other HIV rapid diagnostic tests (RDT), was assessed to be able to guide the development and adoption of the device in Senegal and South Africa. Using convenience sampling, potential participants were recruited from catchment areas where HIV was prevalent. Trained users performed an HIV test on participants using an Oral HIVST. The professional user’s interpretation of results was then measured against the results of various other RDTs. The South African study had 1652 participants and the Senegalese, 500. Most of the participants in each study were 18–35 years old. Senegal had a higher number of females (346/500, 69.2%) compared to South Africa (699/1662, 42.1%). Asante displayed very high sensitivity and specificity when tested against other devices. In the final enzyme-linked immunosorbent assay (ELISA) comparison, in South Africa, the sensitivity: specificity was 99.1:99.9% and in Senegal, 98.4:100.0%. Senegal further identified 53/63 (84.1%) with HIV-1, 8/63 (12.7) with HIV-2 and 2/63 (3.2%) with HIV-1/2 co-infections. Professional or trained users’ interpretations of Asante results correlated strongly to results when using various RDTs, the ELISA assay and Western blot tests, making it a dependable HIV testing instrument.

## 1. Introduction

The World Health Organization’s (WHO, Geneva, Switzerland) “Treat All” operation, initiated in 2015, directs that anyone who is HIV-positive is eligible for treatment without any restrictions [1]. Despite the uptake of this campaign globally, sub-Saharan Africa (SSA) still displays a substantial number of HIV infections and AIDS-related deaths [2]. The WHO’s 90-90-90 goal [3] within SSA has not been completely realized [4]. South Africa, while having achieved the first 90—people knowing their HIV status—has fallen short in the other two areas [5]. Senegal, however, has not reached any of the “90” targets [6]. These realities are particularly alarming, for if the issues regarding awareness of one’s HIV status and linkage to care are not hastened, then it is uncertain whether countries such as South Africa and Senegal will be able to contribute positively to the WHO’s aim of eradicating AIDS by 2030 [7].

Substantial progress has been made in traditional HIV-testing and counseling services (HTS) offered at clinics and through other outreach activities. However, these efforts are still falling short in attaining rapid widespread access to testing due to several weaknesses: long queuing systems, repeat counseling, insufficient innovative targeted interventions, and inability to reach key populations, including men, adolescent girls and young women [8]. With the onset of the COVID-19 pandemic globally in early 2020, HTS efforts have been further hampered as attendance at health facilities for a range of services has declined sharply. To address these barriers to widespread and expansive rapid HIV testing, the WHO approved HIV self-testing (HIVST) as an additional approach. It also clarifies stringent compliance requirements for the production, dissemination, usability, and performance of HIVST devices, in its pre-qualification processes (WHO-PQ) [9].

Evaluations and operational research into the use of HIVST devices by lay individuals are ongoing. Such investigations not only provide information on how to improve HIVST usability as exemplified in a recent South African study [10], but also the acceptability of such devices, as shown in an SSA study [11].

The WHO-PQ Technical Specification Series (TSS) for HIVST requires that candidate products are evaluated across a range of settings and conditions varied by user type. One of these requirements is to evaluate the test performance when conducted by trained professional users. Further, where the test makes any performance claims for the detection of HIV-1 and HIV-2, recruitment sites are needed in areas where these strains dominate. Any in vitro diagnostic manufacturer seeking to gain approval of their HIVST device needs to comply fully with the TSS. In cases where an in vitro diagnostic does not comply, complete justification needs to be provided. The TSS does not measure the clinical utility of the test but focuses on performance characteristics. There are three parts of the TSS: part 1—analytical performance characteristics, which are determined under controlled laboratory conditions; part 2—performance characteristics for professional use; and part 3—performance characteristics in the intended population (i.e., untrained lay individuals) [9].

However, the efficient use of HIVST devices by professional users such as nurses and healthcare workers is also important when one considers that they find themselves responsible for conducting tests to confirm lay individuals’ results, and because, at times, uncertain novice users have indicated that they would like the assistance of professionals in providing further instructions and counseling. This need was highlighted in post-test interviews with lay individuals in a study undertaken in KwaZulu Natal, South Africa [12]. In another investigation in Tanzania, which sought to develop the HIV self-testing experience, some participants indicated their willingness to undergo confirmatory tests from health professionals. This investigation further suggested that education was required in providing direction on how to use HIVST [13]. To direct the successful use of HIVST devices by not only novice users but also by trained users, Ezintsha, through the HIV Self-Testing Africa (HSTAR) program, provides support to HIV test developers by independently evaluating products and gathering relevant data for WHO-PQ submission.

One device, the Asante HIV-1/2 Oral Self-Test (Asante) (Sedia Biosciences, Portland, OR, USA), is a single-use, in vitro diagnostic for the detection of antibodies to HIV-1 and HIV-2 in oral fluid specimens. It is intended for use by professional heath care workers or by lay individuals who have no prior training or experience in performing in vitro diagnostic testing. The test is a new HIV rapid point-of-care test that offers immediate test results and can reduce the burden for routine testing for negative or sero-discordant partners. Thus, it has the potential for a greater number to people to know and sustain the monitoring of their HIV status.

This project, which was undertaken in both South Africa and Senegal to comply with the part 2 requirements of the TSS, investigated how professional users engaged with the Asante device. It aimed to: (i) evaluate the performance of Asante when performed by a professional trained user against the gold standard and national testing platform, (ii) provide the manufacturer with data to be used in the development of the Asante HIVST and (iii) inform submissions for WHO-PQ to the relevant country’s health authorities for approval of sale of the device.

## 2. Materials and Methods

### 2.1. Study Design

This was a controlled study to evaluate the performance of Asante. Using this kit, trained professionals collected samples from participants. The test results were interpreted by the trained user performing the test, and then by a second healthcare professional for confirmation. The results were, thereafter, compared to a professional RDT selected from the tests used by the countries’ National HIV Testing algorithm and a second RDT was performed for cases which were initially interpreted as positive, culminating in a 4th generation enzyme-linked immunosorbent assay (ELISA) laboratory test. Further testing was performed if the ELISA test and HIVST were discordant. Study participants who were deemed confirmed positives for HIV were referred for clinical treatment and care.

### 2.2. Product

In Asante (Figure 1)**,** English was used in South Africa and French in Senegal. Asante is a manual point-of-care, visually read immunoassay for the qualitative detection of antibodies to HIV-1 and HIV-2 in human oral fluid. Results can be obtained in approximately 20 min. The boxed kit contains one of each of the following: pamphlet with instructions for use (IFU) (Appendix A: Figure A1 and Figure A2), test box with tube-holder, pouched test strip, pre-measured sample buffer tube, test strip, oral swab, and desiccant. The test strip is composed of several materials that in combination are capable of detecting HIV antibodies when those antibodies are added to the tube of the sample buffer.

### 2.3. Study Sites

In South Africa, the primary investigation site was that of the HSTAR program located in Hillbrow, central Johannesburg. Additional sites within the HSTAR program in Gauteng, South Africa were also used for recruitment. These chosen sites all possessed conditions and facilities that were conducive to the study and complied with all local government requirements for HIV testing and reporting. In Senegal, lnstitut de Recherche en Sante, de Surveillance Epidemiologique et de Formation (IRESSEF) was the primary study site supplemented by the surrounding public health centers. Here, the prevalence of HIV-2 was evaluated, since it is predominantly found in West African nations [14].

### 2.4. Study Participants and Recruitment

This study recruited anyone who was over the age of 18 and met the inclusion and exclusion expectations. Race, ethnicity, and sexual orientation were disregarded. Recruitment strived to include as close to a 50:50 ratio of males to females as possible.

Various methods were employed to recruit potential participants. South Africa and Senegal recruited prospective participants in a community-based drive. In South Africa, these recruits were from clinical trial sites operating in Hillbrow, and catchment areas in the inner city of Johannesburg. Additionally, recruitment took place using the word-of-mouth approach and from HSTAR-based sites, where enrollment took place on site or participants were brought to the primary study site. In Senegal, people who were accessing health services at public health clinics in the cities of Dakar and Ziguinchor were invited to participate. In addition, both countries recruited candidates who were found to be HIV-positive (and HIV treatment naïve) from other clinical projects run at the same clinical sites.

Before enrolment, potential participants were screened against the inclusion and exclusion criteria. The inclusion criteria were that potential candidates were at least 18 years old, understood and completed the written informed consent form, could perform the testing on the assigned day/s, were willing to impart their medical history and provide samples of oral fluid or blood, and provide a further blood sample by venipuncture if required. Excluded were those who failed to meet the inclusion criteria, had already received any experimental HIV vaccine, or who were at that time on an HIV preexposure prophylaxis regimen, or taking any HIV medication for more than three months. Individuals who refused to use the biometric enrolment system or were unable to provide an identity document or any other document required for enrolment in the said system were also excluded. Furthermore, anyone who displayed any condition which in the facilitator’s opinion would threaten the completion and/or integrity of the investigation—for example, the recruit having an acute illness or showing signs of intoxication—was ruled out.

Participant enrollment followed the serial process outlined in Figure 2.

### 2.5. Sample Size

As per the requirements for WHO PQ [15], recruitment continued until a minimum of 400 HIV positive participants had been detected. These 400 participants made up 24.1% of the total sample of 1662 participants. In total, 10/1662 (0.6%) were excluded on the basis of their viral load, which indicated that the participant was on an antiretroviral therapy (ART) regimen for more than 3 months, and their p24 results.

In Senegal, 500 participants were recruited, and the results of 100% of them were included in the study.

### 2.6. Study Procedures

Once the participant had provided written consent and had been registered, a series of testing steps followed. The participant proceeded into a private room accompanied by the trained user (professional nurse), who performed the Asante self-test. The trained user recorded the result after the specified 20 min run time and a second trained user (professional nurse/healthcare worker), blinded to the first user’s recorded result, read and documented their interpretation.

The primary nurse then performed fingerstick RDTs (RDT 1 First Response™ HIV-1/HIV-2 WB, Product code DD/138, Premier Medical Corporation Ltd., Kachigam, India; RDT 2 ADVANCED QUALITY™ Rapid HIV Test, Product code ITP02002-TC40, InTec Products Inc., Xiamen, China) in accordance with the national HIV testing algorithm.

A 5 mL blood sample was drawn for laboratory gold standard HIV-1 ELISA testing, which was performed on the Abbott Architect 1000SR HIV Ag/Ab combination (Abbott, Green Oaks, IL, USA). Figure 3 further clarifies the procedure that was followed in each country.

In both South Africa and Senegal, the actual testing and confirmatory procedures were identical. However, in the Senegalese study, venous blood was drawn at the onset, while in South Africa it was drawn after the RDTs were performed, and only when required. Furthermore, in South Africa, laboratory tests were performed at appropriately certified local laboratories, and ELISA tests at a South African National Accreditation System-approved good clinical laboratory practice-compliant facility. In Senegal, both laboratory and ELISA tests were performed at the IRESSEF. Figure A3 and Table A1 contain the algorithm and test details, respectively.

### 2.7. Data Management

Data from questionnaires, observations, interviews, test results, and safety forms were collected on paper. Data were captured onto electronic databases within two days of collection/receipt. The project team performed daily quality checks of the data collection forms and databases. If any discrepancies were identified, these were immediately addressed.

### 2.8. Data Analysis

Specific critical and non-critical steps were identified from the new HIV device’s IFU. Critical errors occur when a trained user commits operational errors during the assay that could potentially invalidate the results, such as not contacting the device with the gums, spilling the developer buffer, or terminating the process before completion. Non-critical errors occur when the user performs errors that deviate from the IFU but do not invalidate results, such as performing steps out of order or holding the buffer vial rather than placing it on the table. The successful completion of tasks was evaluated as a percentage of the overall process, with all critical errors identified as potential invalid tests. Failure rates (tests that fail to produce a control line with accurate use) were identified as invalid and reported as a failure rate or percentage of failed tests vs. the total number of tests used in the study.

The study endpoints for this trial were the binomial proportion and the exact 95% confidence interval of the proportion for concordance of participants’ test results obtained by the trained user with the confirmatory test results.

The primary sensitivity population was expected to result from study participants who had positive HIV test results. Study participants of unknown HIV status, with a confirmed positive new HIV device test but false negative or invalid HIV test results, were excluded from the sensitivity analysis population.

The primary specificity population was expected to result from the disposition of study participants who had negative HIV test results. Study participants who had a positive HIV test (indicating the need for confirmatory testing) and a negative HIV test result which was not confirmed (by a second rapid finger prick testing) were excluded from the specificity analysis population.

Sensitivity and specificity calculations are represented in Figure 4.

## 3. Results

### 3.1. Demographics

Within the South African context, 1662 candidates were enrolled in the study. Of these, 963/1662 (57.9%) were male. Just over two-thirds of participants were between 18 and 35 years of age (1154/1662, 69.4%). While most of the participants were South African (1442/1662, 86.8%), 201/1662 (12.1%) were Zimbabwean, and just 19/1662 (1.1%) were from other nationalities.

Senegal’s study sample (n = 500) was smaller than South Africa’s, but unlike South Africa, Senegal had more female than male participants: 346/500 (69.2%) and 154/500 (30.8%), respectively. Just over half the participants were in the 18–35 year age bracket (292/500, 58.4%). In total, 434/500 (96.8%) participants were Senegalese, 23/500 (4.6%) Guinean and 43/500 (8.6%) from other nationalities. A comprehensive record of the demographic data can be found in Table 1.

### 3.2. Discordant Results

The South African investigation revealed 17 samples with discordant results between Sedia oral self-tests and at least one of the HIV RDTs, ELISA, p24 assay and viral load tests, and in Senegal there was 1 discordant result. Ten participants were excluded from the study after the determination of their true sero-status through additional testing inclusive of viral load, p24 and/or antibody testing.

Discordant specimens from nine participants were sent to the laboratory for viral load testing. These samples all resulted in a non-detectable viral load, indicative of HIV-positive participants treated with ART being virally suppressed. The probability of ART viral suppression was indicated in the test results of an additional five participants. These subjects were coded as Asante false negatives due to the absence of any remaining plasma or serum with which to conduct additional viral load testing. The results of one sample were concordantly negative between the Asante test and RDT, but positive by 4th generation ELISA and p24 test. This sample was excluded from the study as it detected p24 antigen positivity and not HIV antibodies.

In Senegal, one subject’s results showed discordance between the Asante Oral self-test and at least one of the HIV RDTs, ELISA, p24 assay and viral load tests. The result of one HIV self-test sample was interpreted as positive by the first trained user, but as “do not know/not sure” by the second reader. The first reader’s interpretation was used in the sensitivity and specificity calculations.

### 3.3. Sensitivity and Specificity

#### 3.3.1. Asante vs. RDT 1

In South Africa, when comparing the Asante self-test results read by Nurse 1 to RDT 1 (First Response™, Premier Medical Corporation Ltd., Kachigam, India), there were two cases of discordancy (one false negative and one false positive). No critical errors were reported. There were 400/1652 (24.2%) RDT 1 HIV-positive readings and 1252/1652 (75.8%) HIV-negative readings. The sensitivity was 99.75% and specificity 99.92%. In Senegal, the self-test comparison to RDT 1 (SD Bioline HIV-1/2, Standard Diagnostics Inc., Yongin-si, Korea) showed no discordancy. The sensitivity was 100%, as was the specificity (Table 2).

#### 3.3.2. Asante vs. RDT 2 and RDT 3

The next confirmatory test (Advanced Quality rapid test, Intec Products Inc., Xiamen, China) was only performed on the positive cases that resulted from the HIVST and or RDT 1. The total number tested in South Africa, therefore, was 401/1652 (24.2%), of which 1/401 (0.24%) tested HIV-negative. The sensitivity was 99.7% and the specificity was undetermined. Senegal, unlike South Africa, used two additional RDTs (RDT 2, which was SD Bioline HIV-1/2, Standard Diagnostics Inc., Yongin-si, Korea, and RDT 3, which was Multisure HIV-1/2 Rapid Test, MP Biomedicals, CA, USA) against which to confirm the positive readings. Comparisons to both RDT 2 and RDT 3 indicated a 100% sensitivity and 100% specificity. (Table 2). A Western blot test was thereafter used on the HIV-positive samples and there were found to be 53 HIV-1 only, 8 HIV-2 only, and 2 HIV-1/2 co-infections.

#### 3.3.3. Asante vs. ELISA

In performing the confirmatory test against ELISA, the entire sample in each country was used (1652 in South Africa and 500 in Senegal). South African tests showed a total of 7/1652 (0.4%) cases of discordancy (2 false positives and 5 false negatives) between the Asante self-test and 4th generation ELISA. This yielded a sensitivity of 98.8% and a specificity of 99.8%. Senegal recorded one case of discordancy (zero false positive and one false negative). When comparing the HIVST result with the laboratory-based ELISA confirmatory test, there was one case of discordancy (zero false positive and one false negative). That specimen was additionally tested using the Western blot test and was determined to be reactive to only p24. Therefore, it was excluded from the analysis of sensitivity, resulting in the total valid subjects being 499. The resultant sensitivity was 98.4% and specificity 100% (Table 2).

#### 3.3.4. First Trained User’s Reading versus Second Trained User’s

This comparison of the first trained user’s reading of the Asante HIVST to that of a blind reading by a second trained professional showed no discordancy in the reading, thereby yielding a sensitivity as well as a specificity of 100%. The reading was based on 1652 samples that were used at the beginning of the testing process. This comparative analysis was performed in South Africa only.

#### 3.3.5. RDT 1 vs. ELISA

South Africa conducted another additional comparison: RDT1 vs. ELISA assay. From the sample of 1652 participants, there were a total of five cases of discordancy (one false positive and four false negatives). The sensitivity that resulted was 99.8% and specificity was 99.9%.

As determined by the Western blot test, the results identified 53/63 (84.1%) with HIV-1, 8/63 (12.7%) with HIV-2 and 2/63 (3.2%) with HIV-1/2 co-infections.

## 4. Discussion

This is the first report of the performance assessment of the Asante self-tests being used by a professional or trained user. Adding additional significance to the study is the fact that it was conducted in both South Africa and Senegal independently with the same protocol. The sensitivity and specificity in both the South African and Senegalese studies were extremely high. Despite differences in the study site environments, the Asante tests held up well in comparison to other RDTs that were used in the study, as well as the ELISA assay. With regards to the validity of the HIVST results, a recent South African study displayed similar findings in their assessment of a different oral HIVST [17], where sensitivity was 99.2% and specificity was 100%.

What has emerged is that Asante HIVSTs are accurate in detecting HIV when used correctly. Neither critical errors nor invalid results were reported. While the investigation did yield a few discordant results, which mostly arose from participants who were likely to already be on ART or who tested positive for the p24 antigen, when Asante was compared to other RDTs and ELISA, these were minimal in comparison to the sample.

Given that HIV has global presence and is high in Africa particularly [18], the almost 100% sensitivity and specificity results yielded by Asante bodes well for professional users who can use it to conduct reliable initial HIV test or even provide confirmation of a self-tester’s interpretation of their results. These results support the use of additional training materials and client support when performing an HIVST.

The necessity for dependable data is underscored in a study that focused on the tracking and controlling of HIV in some at-risk populations in Africa. The investigation explains how data inform funding and government policies regarding HIV testing and treatment [19]. The negative impact of reporting inaccurate HIV data is further elaborated upon by Etoori [20], where poorly trained staff is pinpointed as one of the areas that needs revision in South Africa. This means that the Asante self-testing device has value as it provides high accuracy with easy device reading and interpretation for the professional user.

Engel et al. [21] acknowledged the multitude of problems associated with testing for diseases in different circumstances, and emphasized that irrespective of the problem, the main factors contributing to post-test care are the communication between patients and the healthcare worker, as well as communication between health professionals themselves. One of the identified reasons for breakdown in communication was the frustration linked to the time it took for processes to be completed. HIVSTs can reduce the time an individual may have to wait between visiting a healthcare worker for testing and receiving further testing and counseling, as is iterated by Maheu-Giroux and Tanser [22]. In their investigation on linkage to care in community-based organizations, Shamu and Slabbert [23] further explained the additional need for HIV testing and counseling to extend beyond health facilities, to not only ease the burden of duty upon healthcare workers such as nurses, but to also to expedite HIV-related testing and counseling. A 2015 study on the uptake of HIV testing in Africa concluded that HIVST devices that were in line with the WHO normative guidance could be used as an effective supplement to traditional testing. Their research also indicates that respondents from all three countries, who were part of the study, welcomed the distribution of HIVST from health facilities beyond clinics, such as pharmacies [24]. It appears, therefore, that the Asante HIVST could well be a solution to promote the uptake of HIV testing.

There are currently limited options for pre-qualified oral fluid HIVSTs, with OraQuick^®^ (OraSure Technologies, Bethlehem, PA, USA) [25] being the most-used test. There is a need to evaluate other candidate oral fluid HIVSTs, both for improving availability and stimulating competitive (lower) pricing. This evaluation of Asante HIVST offers data for pre-qualification, and if successful, possible introduction of another product to the HIVST market.

### Limitations

This study reported on the accuracy of trained users’ interpretations of results when using the Asante self-test in comparison to results of confirmatory testing. We did not note the extent to which the “trained user” was schooled in the engagement with the device—particularly in Senegal. This detail needs to be considered, as it is not guaranteed that trained users, irrespective of the level of training and context, will be able to use the device as accurately as those in this controlled study.

Furthermore, the viral load testing of some participants in the South African study could not be performed, as there was insufficient plasma serum with which to complete the test. Despite the ELISA readings for all of these being reactive, these participants’ results were recorded as false negatives, as indicated in the Asante and RDT 1 tests, possibly slightly skewing the performance results.

Finally, convenience sampling was used in this study, limiting its extrapolation to other contexts.

## 5. Conclusions

It was established that professional/trained users’ interpretations of results when using the Asante self-test in both Senegal and South Africa correlated strongly to results of the same samples when using various RDTs, the ELISA assay and a Western blot test. This suggests that the Asante HIVST is a reliable tool with which to test for HIV. It further has the potential to accelerate the rate at which HIV tests are conducted, results are verified, and linkage to care proceeds. The varied verification procedures employed in this study can also be used as a benchmark in the competence testing of similar products for not only users of the product, but also its developers and health authorities.

## Figures and Tables

**Figure 1 diagnostics-11-01727-f001:**
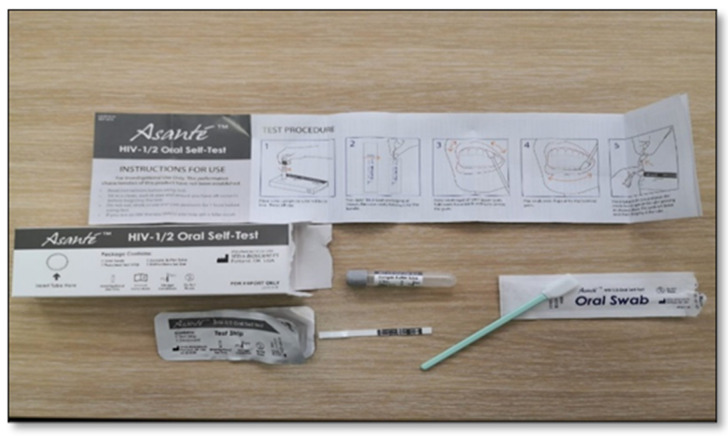
Asante.

**Figure 2 diagnostics-11-01727-f002:**
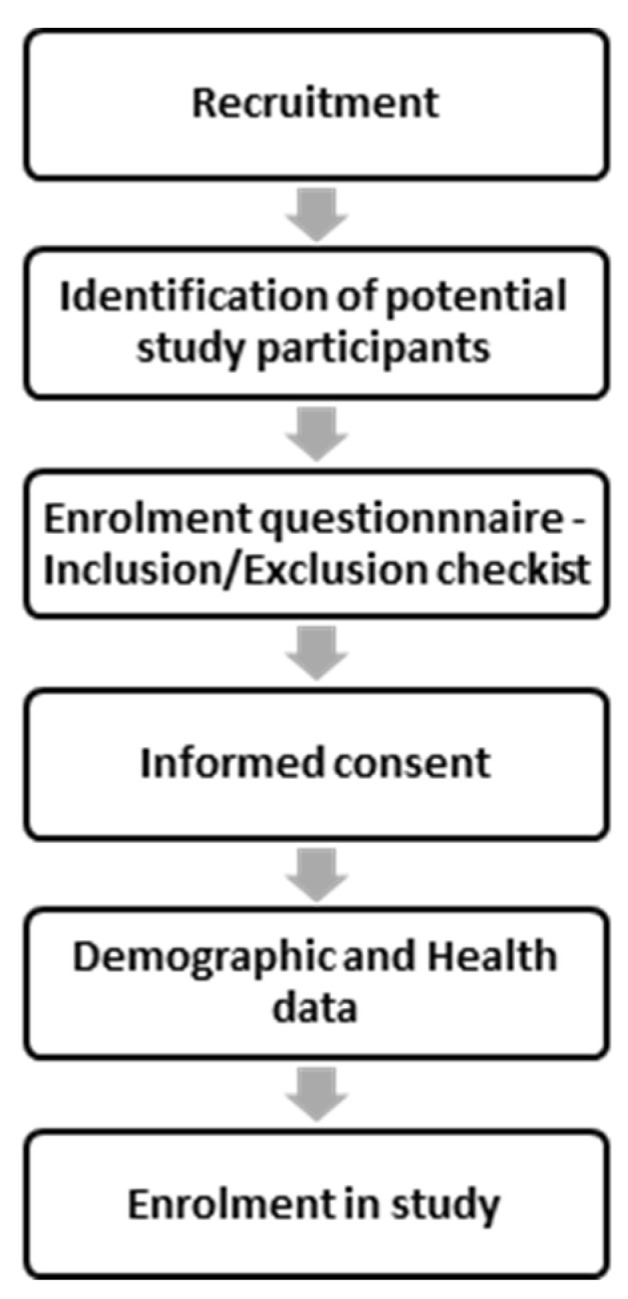
Recruitment and enrolment workflow.

**Figure 3 diagnostics-11-01727-f003:**
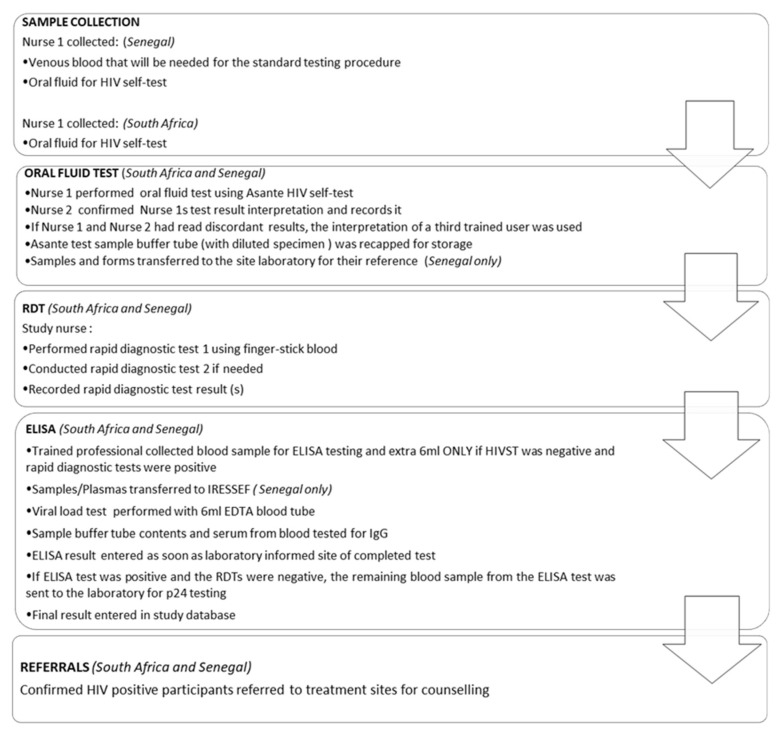
Study workflow.

**Figure 4 diagnostics-11-01727-f004:**
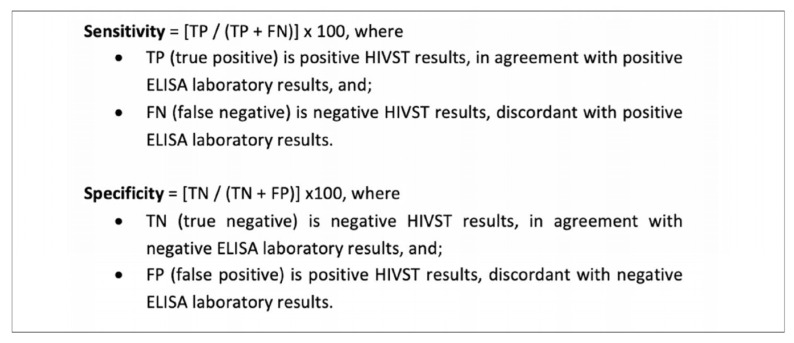
Sensitivity and specificity calculations [16].

**Table 1 diagnostics-11-01727-t001:** Study participant demographics.

Characteristic			South Africa		Senegal	
			Number (n = 1662)	%	Number (n = 500)	%
Gender	Male		963	57.9	154	30.8
	Female		699	42.1	346	69.2
Age	18–25		604	36.3	99	19.8
	26–35		550	33.1	193	38.6
	36–45		339	20.4	110	22.0
	46+		169	10.2	98	19.6
Nationality	South Arica	South Africa	1442	86.8		
		Zimbabwe	201	12.1		
		Other	19	1.1		
	Senegal	Senegal			434	96.8
		Guinea			23	4.6
		Other			43	8.6

**Table 2 diagnostics-11-01727-t002:** Sensitivity and specificity of Sedia HIV-1/2 Oral self-tests vs. RDT 1, sensitivity and specificity of Sedia HIV-1/2 Oral self-tests vs. RDT 2, sensitivity and specificity of Sedia HIV-1/2 Oral self-test vs. RDT 3 (Senegal only), sensitivity and specificity of HIV-1/2 Oral self-tests vs. ELISA.

**Sedia HIV-1/2 Oral Self-Tests**	**RDT 1**							
	**South Africa (n = 1652)**				**Senegal (n = 500)**			
	**Negative** **n (%)**	**Positive n (%)**	**Indeterminate n (%)**	**Total**	**Negative** **n (%)**	**Positive** **n (%)**	**Indeterminate** **n (%)**	**Total**
Negative	1251(99.9)	1 (0.0)	0 (0.0)	1252	437 (100.0)	0 (0.0)	0 (0.0)	437
Positive	1 (0.2)	399 (99.8)	0 (0.0)	400	0 (0.0)	63 (100.0)	0 (0.0)	63
Total	1252 (75.8)	400 (24.2)	0 (0.0)	1652	437 (87.4)	63 (12.6)	0 (0.0)	500
Sensitivity	99.8%				100%			
Specificity	99.9				100%			
**Sedia HIV-1/2 Oral self-tests**	**RDT 2**							
	**South Africa (n = 401)**				**Senegal (n = 63)**			
	**Negative**	**Positive**	**Indeterminate**	**Total**	**Negative**	**Positive**	**Indeterminate**	**Total**
Negative	N/A	1 (100.0)	0 (0.0)	1	N/A	0 (0.0)	0 (0.0)	N/A
Positive	1 (0.2)	399 (99.8)	0 (0.0)	400	0 (0.0)	63 (100.0)	0 (0.0)	63
Total	1 (0.2)	400 (99.8)	0 (0.0)	401	0 (0.0)	63 (100.0)	0 (0.0)	63
Sensitivity	99.7%				100%			
Specificity	Insufficient for calculation				100%			
**Sedia HIV-1/2 Oral self-test**	**RDT 3**							
	**Senegal (n = 500)**							
	**Negative**		**Positive**		**Indeterminate**		**Total**	
Negative	N/A		0 (0.0)		0 (0.0)		N/A	
Positive	0 (0.0)		63 (100.0)		0 (0.0)		63	
Total	0 (0.0)		63 (100.0)		0 (0.0)		63	
Sensitivity	100%							
Specificity	100%							
**Sedia HIV-1/2 Oral self-tests**	**ELISA confirmatory test**							
	**South Africa (n = 1652)**				**Senegal (n = 499)**			
	**Negative**		**Positive**	**Total**	**Negative**		**Positive**	**Total**
Negative	1247		5	1252	436		0 (0.0)	436
Positive	2		398	400	0 (0.0)		63	63
Total	1249		403	1652	436		63	499
Sensitivity	98.8%				98.4%			
Specificity	99.8%				100%			

Note: Percentages calculated using row totals.

## Data Availability

The data presented in this study are available on request from the corresponding author. The data are not publicly available to maintain participant confidentiality.

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
