# Peer review of "Usability and Clinical Performance Characteristics of the Asante HIV1/2 Test by Trained Users in Two African Sites"

_diagnostics, 2021, doi:10.3390/diagnostics11091727_

Round 1
Reviewer 1 Report
Majam et.al. tested the performance of (Asante) HIV-1/2 Oral Self-Test and compared it to other RDTs, and conventional ELISA, in two study cohorts; one in Senegal, and the other is in South Africa.
The test was found to have a very high sensitivity and specificity in detecting HIV antibodies. The study is well presented, and the findings and conclusion is well supported by the results.
Comments:
- There are multiple spelling and grammatical mistakes in the manuscript, and I recommend correcting them; such as “focuses” in line 67, layperson instead of lay individual, the extra brackets in line 261..etc. Also the proper use of commas and semicolons is advised to enhance the readability of the manuscript.
- In line 129: „Race, gender, ethnicity, and sexual orientation were disregarded” How was gender and ethnicity disregarded? If I understand correctly this was controlled for in the analysis, and the results were stratified by gender and ethnicity.
- In line 159: 24% of 1662 is almost 399, also, the total number of participants was 1662, and 10 were excluded? But 1658 completed the study? Wouldn’t it be better to say 1658 completed the study and 4 were excluded? It sounds a little confusing.
- In line 171: how could the professional nurse be blinded to the results? Didn’t he/she had a chance to interpret the result of the RDT read by the first nurse? and thereafter document the interpretation? As mentioned in figure 3?
- The authors mention that “The focus of this study was to assess the accuracy of trained users’ interpretations of 353 results when using the Asante self-test in comparison to results of confirmatory testing”, however, there is very little information to support that claim, as the study was majorly focused on assessing the performance of the test against other RDTs. If indeed that was the main focus, then it would make more sense to analyze this interpretation in detail, and also comparing it to interpretation by participants. I therefore recommend either supporting this focus by other findings, or denoting it as secondary objective.
- Providing a percentage in table 2a/b between brackets would be very useful.
Question:
- How does the test perform in analyzing individuals who have recently seroconverted? Was this explored? It would be highly informative if sensitivity could be established in early infection.
In conclusion, I find the study well suited for the journal, and informative for those involved in the field of HIV testing. However, addressing the comments above is essential.
Author Response
Review report 1
English language and style
(x) English language and style are fine/minor spell check required
Thank you. The manuscript has been read through again and revised/corrected where necessary.
Comments and Suggestions for Authors
Majam et.al. tested the performance of (Asante) HIV-1/2 Oral Self-Test and compared it to other RDTs, and conventional ELISA, in two study cohorts; one in Senegal, and the other is in South Africa.
The test was found to have a very high sensitivity and specificity in detecting HIV antibodies. The study is well presented, and the findings and conclusion is well supported by the results.
Comments:
- There are multiple spelling and grammatical mistakes in the manuscript, and I recommend correcting them, such as “focuses” in line 67, layperson instead of lay individual, the extra brackets in line 261 etc. Also, the proper use of commas and semicolons is advised to enhance the readability of the manuscript.
The manuscript has been read through again and revised/corrected where necessary.
- In line 129: “Race, gender, ethnicity, and sexual orientation were disregarded” How was gender and ethnicity disregarded? If I understand correctly this was controlled for in the analysis, and the results were stratified by gender and ethnicity.
The aim of the paper was to assess usability and performance therefore demographic characteristics were not inclusion criteria. We only report the sex and ethnicity to give a description of the participants. Recruiters did however try to get an equal sex disaggregation, so the sentence has been amended to
“Race, ethnicity, and sexual orientation were disregarded”
- In line 159: 24% of 1662 is almost 399, also, the total number of participants was 1662, and 10 were excluded? But 1658 completed the study? Wouldn’t it be better to say 1658 completed the study and 4 were excluded? It sounds a little confusing.
24% was edited to 24.1%. As 1658 is only reported once and does not provide context for the 1552 denominator reported on several times in the manuscript we have deleted the phrase “1658/1662 (99.7%) completed the study,”
- In line 171: how could the professional nurse be blinded to the results? Didn’t he/she had a chance to interpret the result of the RDT read by the first nurse? and thereafter document the interpretation? As mentioned in figure 3?
Nurse 1 reads and records the results. Nurse 2 independently reads and records the results and then they compare. The phrase on blinding has been clarified.
“The trained user recorded the result after the specified 20-minute run time and a second trained user (professional nurse/healthcare worker), blinded to the first user’s recorded result, read, and documented their interpretation.”
- The authors mention that “The focus of this study was to assess the accuracy of trained users’ interpretations of 353 results when using the Asante self-test in comparison to results of confirmatory testing”, however, there is very little information to support that claim, as the study was majorly focused on assessing the performance of the test against other RDTs. If indeed that was the main focus, then it would make more sense to analyze this interpretation in detail, and also comparing it to interpretation by participants. I therefore recommend either supporting this focus by other findings or denoting it as secondary objective.
We note this point and have edited the limitations section to remove confusion about the focus of the study.
- Providing a percentage in table 2a/b between brackets would be very useful.
Percentages have been added to tables 2 a-d.
Question:
- How does the test perform in analyzing individuals who have recently seroconverted? Was this explored? It would be highly informative if sensitivity could be established in early infection.
This was not explored as part of this device evaluation for prequalification. Sedia has another device aimed at recency and will likely commence evaluation for prequalification by the end of 2021. We agree that early detection would greatly improve the testing programme and anticipate that it will be a greater product evaluation focus as more tests become available on the market.
In conclusion, I find the study well suited for the journal, and informative for those involved in the field of HIV testing. However, addressing the comments above is essential.
Reviewer 2 Report
The study presented in the manuscript is extremely important for the approval and validation of an HIV self-testing device that intends to be used in Senegal and South Africa to identify HIV infected persons in order to refer them for clinical treatment and care. The findings of this study are very important for the diagnosis of HIV infected population as the WHO goal is to eradicate the virus till 2030 and still, in the African countries this goal is far from achieved.
More information should be added regarding the method used in ASANTE HIV-1/2 Oral Self-Test for determining the HIV-1 and HIV-2 antibodies.
The quality of Figure A2 is very low and it can not understand the text.
In discussion, it is important to emphasize why this test brings new compared with other found already on the market.
Author Response
Review report 2
(x) I don't feel qualified to judge about the English language and style
Thank you. Although you have selected this option, we want to highlight that the manuscript has been read through again and revised/corrected where necessary – as per Reviewer 1’s comment.
Comments and Suggestions for Authors
The study presented in the manuscript is extremely important for the approval and validation of an HIV self-testing device that intends to be used in Senegal and South Africa to identify HIV infected persons in order to refer them for clinical treatment and care. The findings of this study are very important for the diagnosis of HIV infected population as the WHO goal is to eradicate the virus till 2030 and still, in the African countries this goal is far from achieved.
- More information should be added regarding the method used in ASANTE HIV-1/2 Oral Self-Test for determining the HIV-1 and HIV-2 antibodies.
We have added more information about the test kit under the Product section.
“Asante (figure 1), English was used in South Africa and French in Senegal. Asante is a manual point-of-care, visually read immunoassay for the qualitative detection of antibodies to HIV-1 and HIV-2 in human oral fluid. Results can be obtained in approximately 20 minutes. The boxed kit contains one of each of the following: pamphlet with instructions for use (IFU) (figures A1 and A2), test box with tube-holder, pouched test strip, pre-measured sample buffer tube, test strip, oral swab, and desiccant. The test strip is composed of several materials which in combination are capable of detecting HIV antibodies when those antibodies are added to the tube of sample buffer.”
- The quality of Figure A2 is very low and it cannot understand the text.
A high-resolution copy has been included in the manuscript.
- In discussion, it is important to emphasize why this test brings new compared with other found already on the market.
The following has been added to the discussion.
There are currently limited options for pre-qualified oral fluid HIVSTs, with OraQuick® [25] being the mostly used test. There is a need to evaluate other candidate oral fluid HIVSTs both for improving availability and competitive (lower) pricing. This evaluation of Asante HIVST offers data for pre-qualification and if approved possible introduction of another product to the HIVST market.
Round 2
Reviewer 2 Report
The authors addressed all my comments. The manuscript is much improved in its current form and is ready for acceptance.